# Sex Differences in Chronic Obstructive Pulmonary Disease: Implications for Pathogenesis, Diagnosis, and Treatment

**DOI:** 10.3390/ijms26062747

**Published:** 2025-03-18

**Authors:** Paulina Czarnota, Jamie L. MacLeod, Niya Gupta, Ani Manichaikul, Yun M. Shim

**Affiliations:** 1Department of Genome Sciences, University of Virginia School of Medicine, Charlottesville, VA 22908, USA; bsk8xx@virginia.edu; 2Department of Medicine, Division of Pulmonary and Critical Care Medicine, University of Virginia School of Medicine, Charlottesville, VA 22908, USA; jlm6re@uvahealth.org (J.L.M.); xgc6yy@virginia.edu (N.G.)

**Keywords:** chronic obstructive lung disease, chronic bronchitis, emphysema, sex difference, pathogenesis, diagnosis, treatment

## Abstract

Chronic obstructive pulmonary disease (COPD) is a leading chronic disease worldwide, with significant healthcare utilization, morbidity, and mortality. Irreversible airflow obstruction identified on spirometry establishes the diagnosis of COPD, but the disease entity encompasses a heterogeneous collection of lung diseases, including chronic bronchitis and emphysema. Despite the enormous burden of COPD, there are no pharmacological therapies that slow its progression or reduce mortality, indicating the need for a deeper understanding. There are sex differences concerning COPD prevalence, pathology, and symptoms. Historically thought to primarily affect males, its effect on females has increased significantly over time due to a rising prevalence of smoking and exposure to harmful pollutants among females. Over the past decade, the age-adjusted prevalence of COPD has been consistently higher in females than in males. Despite this, the impacts of biological sex continue to be confusing and poorly defined. The primary goal of this review is to organize and collate sex-dependent factors that may contribute to disease differences in males and females, thereby identifying future research questions in this area.

## 1. Background: COPD—Definition, Symptoms, and Diagnosis

Chronic obstructive pulmonary disease (COPD) is a collection of heterogeneous respiratory diseases that affect the airways, alveoli, or pulmonary vasculature, leading to chronic respiratory symptoms [1,2]. The Global Initiative for Obstructive Lung Disease (GOLD) 2024 defines COPD as a post-bronchodilator forced expiratory volume in 1 s to forced vital capacity (FEV1/FVC) ratio of less than 0.70, which indicates irreversible airflow obstruction [1]. COPD symptoms vary depending on the location of the respiratory pathology. One common symptom is breathlessness or dyspnea during activities [1,3]. Other symptoms include a persistent cough, recurrent wheezing, and lower respiratory tract infections [1,4]. Establishing a diagnosis of COPD can be challenging because the usual symptoms can be non-specific and overlap with other conditions [5]. Also, comorbid conditions with similar symptoms and risk factors may be present. For example, chronic coughing is a common COPD symptom but can also be caused by asthma, gastroesophageal reflux, or other etiologies [1]. The GOLD 2024 report highlights the COPD Assessment Test (CAT) and St George’s Respiratory Questionnaire (SGRQ) as tools to assess symptoms [1,6,7]. The current GOLD 2024 recommendations define severe COPD symptoms as a CAT score greater than or equal to ten or an SGRQ score greater than or equal to twenty-five [1]. The GOLD 2024 report updated the COPD stratification strategy to the ABE assessment tool. The GOLD 2024 ABE assessment tool stratifies patients based on CAT or Modified Medical Research Council (mMRC) questionnaire scores and the frequency of acute COPD exacerbation to assign the patients to groups A, B, or E [1,8]. The ABE assessment helps to guide clinical decision-making and disease management and considers heterogeneous characteristics of COPD. Yet, the effects of sex differences have not been systematically addressed. Therefore, this review will explore COPD risk factors, disease pathogenesis, and their connections to biological sex. 

## 2. Methods

The main objective of this scoping review is to explore sex differences in three main areas of COPD: 1) disease characteristics and phenotypes, 2) risk factors that may impact the disease development, and 3) disease management. An overview of the steps undertaken to identify relevant studies is presented in Figure 1.

Publications from the past 20 years were reviewed. Search engines including Google Scholar, Web of Science, Embase, and PubMed were used with the keywords “COPD” and “sex differences”. Additional filtering keywords were then used to refine our search, which included but was not limited to “emphysema”, “chronic bronchitis”, “chronic obstructive pulmonary disease”, “smoking”, “man”, “woman, “female”, “male”, “management”, “symptoms”, “aging”, “sex hormones”, “biomass”, “BMI”, and “air pollution”. Reported sex differences related to COPD were quite variable in the manuscripts identified using this process. Therefore, each manuscript required an in-depth review to determine if the results were of interest for this review. This manual review required us to include phrases such as male/female and men/women followed by sex-stratified methods, results, and discussions. Moreover, the manuscripts whose primary focus was not COPD were removed unless necessary for a thorough background review. After searching, reviewing, and filtering, 80 manuscripts were selected, collated by the subtopics, and reviewed.

## 3. Results

### 3.1. COPD Phenotypes: Emphysema and Chronic Bronchitis

COPD is historically categorized into two phenotypes: chronic bronchitis and emphysema. Chronic bronchitis is defined by “coughing and sputum production for more than three months in at least two consecutive years” [9]. Emphysema is characterized by a loss of elasticity in the gas exchange units of terminal airways and alveoli, accompanied by hyperinflation of the lungs [10,11]. Recent research has investigated whether different phenotypes are more frequent in either sex. Emphysema has been reported to occur more extensively in males than in females when assessed by emphysema extent in chest CT (Table 1) [12,13,14,15]. These differences persist across GOLD stages [12,14] and even after accounting for smoke exposure [13]. The smoking habits in individuals with emphysema vary significantly by sex. Females tend to display lower cumulative smoke exposure than males [12,13]. However, among patients with early-onset COPD, the severity of emphysema appears similar between males and females [12]. These findings reflect complex interactions between sex differences and cigarette smoke exposure during the development of emphysema. Emphysema can be further subtyped as panlobular or centrilobular subtypes [16]. In panlobular emphysema, the entire lobules are affected, while in centriobular emphysema, more central areas of lobules are affected [16]. A study by Wilgus et al. [16] demonstrated no statistically significant sex differences in the prevalence between emphysema subtypes. Historical studies reported that males developed more extensive emphysematous changes [12,13,14,15], but additional studies are needed to examine sex differences. Many studies have investigated the prevalence of chronic bronchitis between sexes, but there remains no consensus. A study from South Korea reported a higher incidence of biomass exposure and bronchiectasis in females than in males (Table 1) [17]. The Danish Twin Registry Study reported a stronger association between the diagnosis of chronic bronchitis and the female sex (Table 1) [18]. However, another study from France contradicted the results from the South Korean study, reporting that males had a higher incidence of chronic bronchitis with an increased risk of undiagnosed chronic bronchitis (Table 1) [19]. Other studies reported females as being more prone to exhibiting a small airway disease phenotype in COPD and males as being more prone to exhibiting emphysema (Table 1) [15,20]. While the natural history of chronic bronchitis has been explored in some studies, these were often small studies in isolated geographic locations, limiting their generalizability. Therefore, a significant need persists for wider-scale global studies to determine the natural history of emphysema and chronic bronchitis stratified by sex differences.

### 3.2. COPD Risk Factors

A variety of factors contribute to the prevalence and development of COPD. This section will focus on COPD risk factors and the variable effects of these risk factors based on sex.

#### 3.2.1. Cigarette Smoking

Cigarette smoking is the most important risk factor for COPD. Ever-smokers have a higher COPD incidence compared to never-smokers, and cumulative smoke exposure, as measured by pack-years of smoking, is one of the strongest associations [45]. Females appear to have a similar or even higher risk of developing COPD, as they exhibit anatomic vulnerabilities to cigarette smoke, such as smaller lungs and divergent smoke metabolism compared to males (Table 1) [21,22]. Female smokers exhibit more airflow obstruction than their male counterparts when adjusted for cumulative smoke dosage [22]. However, other reported effects of cigarette smoke exposure are divergent. Female smokers with early-onset COPD are reported to have a lower cumulative smoke exposure than older females with comparable COPD severity [46]. In a study by Hardin et al., sex-specific differences in COPD and sex-specific associations between smoking and emphysema were investigated [12]. Among patients with early-onset COPD, severe emphysema, and GOLD grade IV COPD, females had comparable radiographic emphysema to males, despite having fewer smoking pack-years, controlling for covariates such as pack-years of smoking, age, and BMI [12]. However, among all subjects who were ever-smokers, males had worse emphysema than females [12]. Regional variations, including socioeconomic factors, smoking patterns, and noncigarette environmental exposures, will need to be considered in future studies to determine sex differences in larger global populations. 

#### 3.2.2. Biomass Fuel Exposure

Biomass fuel exposure has been evaluated among females from developing countries such as India or China. Indoor pollutants with inadequate ventilation pose a significant risk in the development of COPD among females (Table 1) [23,24]. In a study by Sana et al., exposure to biomass smoke significantly increased the risk of developing COPD among females living in urban and rural areas [23]. This finding was further supported by a study conducted in China in which females from rural areas had a high incidence of COPD [24]. These reports emphasize the detrimental impact that limited household ventilation can have, causing higher indoor biomass exposure and therefore a higher occurrence of COPD in females, especially in rural areas [23,24]. Generally, studies focused on examining biomass fuel exposure in the context of COPD development have also been conducted in populations with low cigarette smoke exposure. By virtue of this, the confounding effects of cigarette smoke exposure are reduced in these studies. Female COPD prevalence in India was found to be too high to be solely attributed to cigarette smoking, as a relatively small fraction of females in India smoke cigarettes [22,47]. Similar observations have been reported in other parts of the world [48]. Females in many cultures are primarily responsible for domestic chores. Still, few studies have evaluated sex differences in COPD prevalence related to domestic biomass smoke exposure. To date, the emphasis has been on females from rural regions of developing countries. A study by Zhang et al. from China observed that females exposed to biomass smoke were at a much higher risk of developing COPD than their male counterparts [24]. These findings highlight that biomass smoke exposure affects females more negatively than males. This underscores the need for targeted interventions to improve indoor ventilation where reliance on biomass fuel is common. Studies focused on biomass fuel exposure would also benefit from examining sex differences in the context of this risky environment.

#### 3.2.3. Non-Occupational and Occupational Exposures

Air pollution is a major contributor to overall morbidity and mortality worldwide and has been speculated to increase the risk of developing COPD [49,50]. Air pollution can be defined as gaseous (such as nitrogen dioxide and ozone) or particulate (usually particles below the size of 10 μm or smaller [PM10], such as smoke particles). Air pollutants have been associated with acute exacerbations of COPD (AE-COPD) by exacerbating airway inflammation [51], and PM10 has been associated with a higher risk of AE-COPD (Table 1) [26]. Several studies have assessed sex differences connected to AE-COPD and air pollution exposure, without a clear consensus. A study by Liang et al. (Table 1) [27] reported a higher risk of air-pollution-correlated AE-COPD in females older than 65 years of age compared to males, while a study by Jung et al. [26] reported an association between the male sex and PM10 exposure in AE-COPD. Another factor to consider in COPD risk is the difference in occupational hazards and workplace exposure to fumes and PM pollutants. COPD risk was found to be correlated with occupational hazards for certain job groups, such as miners (Table 1) [25]. Between males and females, workplaces and industries burdened by occupational smoke hazards exhibited a correlation with COPD incidence before and after retirement [25]. While occupational and non-occupational smoke exposure is known to be related to COPD risk and development, little research has been conducted to address the sex differences in that context. Moreover, studies need to focus on sex differences in exposure to particulate matter to examine which sex may be more susceptible.

#### 3.2.4. Body Mass Index (BMI)

Low BMI in early adult life is linked to increased odds of COPD-related mortality [52,53]. Patients in different BMI strata exhibit varying disease characteristics and adverse outcomes related to COPD [53]. In a study by Grigsby et al., the highest COPD prevalence was observed in the group of underweight patients, and those with a low BMI (<19.8 kg/m^2^) were reported to have higher odds of developing COPD than groups with higher BMI (Table 1) [28]. The association between weight loss, low BMI, and decreased rates of survival in COPD was also reported when healthy and COPD-affected male populations were compared [52]. A study by Wada et al. noted that weight loss of “more than 10 kilograms since age 20 and BMI below 18.5 kg/m^2^” contributed to a higher COPD mortality risk [52]. It is worth noting that females were not included in Wada et al.’s study [52], so sex differences could not be evaluated. In another study, the “underweight” category of BMI < 18.5 kg/m^2^ was characterized by a more significant proportion of COPD cases among males than females [28]. Males with diagnosed COPD have been reported as being more likely to experience a slower decline in lung function if they are overweight or obese [54]. In contrast, little evidence of a similar trend has been demonstrated in females [54]. 

#### 3.2.5. Aging and Associated Physiological Changes

Aging is a recognized risk factor for COPD and a contributor to its progression [55,56]. Lung function progressively declines with aging, starting from the early 20s [56]. From the perspective of the respiratory system, aging is usually associated with the progressive loss of elasticity in the lung, mimicking the phenotype of mild emphysema (so-called “senile emphysema”) [57]. Aging is also accompanied by the weakening of the respiratory muscles and reduced chest wall compliance, which induces increased air trapping [56]. Atrophy of respiratory muscles and less effective mucociliary clearance are major contributors to higher susceptibility to COPD in elderly individuals due to the inflammatory milieu from ineffective cough and mucus removal [57]. Smoking status is a prominent factor contributing to airflow obstruction and pulmonary function decline, which needs to be considered in age- and sex-dependent COPD pathogenesis. Both Han et al. [58] and Brandsma et al. [57] in their literature reviews highlighted the findings of more rapid lung function decline in females compared to their male counterparts [58] and more rapid decline in males in the general population [57]. Those studies, however, do not highlight how factors such as BMI, smoking status, cumulative smoke exposure, or lung size were considered. Whether sex exhibits a more significant age-related pulmonary function decline was examined in a 2024 study by Sangani et al. (Table 1) [29]. They identified that the absolute decline in FEV1 per year was more pronounced in males than females with age- and height-stratified analyses. In contrast, no sex differences were noted for the relative changes in FEV1 over 6 years [29]. A recent systematic review by Adeloye et al. (Table 1) [30] reported that COPD prevalence and risk were higher in males globally. Even though these studies highlight sex differences in COPD in the context of age, they rarely focus on the biological and molecular aspects of age and sex influencing susceptibility to COPD. Research on cellular senescence in aging is limited to mouse models of COPD/emphysema [59]. This highlights the need for studies focused on mechanisms of aging and how it affects sex differences in COPD in humans. Aging is also correlated with fluctuations in sex hormones, which can impact the natural history of COPD. For now, animal models provide insight into the connection among sex hormones, lung function, and lung pathology. Murine studies reveal less pronounced changes in lung morphology in the small airways of females after oophorectomy, suggesting that sex hormones may play a significant role [60]. Female mice also had impaired respiratory antioxidative processes, but the impairment was subsequently lost after oophorectomy [60]. Several studies suggest the possibility that sex hormones affect the prevalence of COPD phenotypes [15,20]. These findings suggest that sex hormones and their impact on COPD need to be further evaluated in humans.

## 4. COPD and Sex Differences in Pulmonary Anatomy

### 4.1. Respiratory Anatomy

Individual lung size and volume vary and are known to be affected by height [61]. A study by Dominelli et al. reported that even after adjusting for height as a confounder, the airways of males have larger luminal areas than females by up to 20–30% [61]. The never-smoker group from the COPDGene study showed that females had a lower total airway count and airway fractal dimensions, smaller diameter of the segmental airways, and lower lumen parameters of airway volume than males [13]. Anatomical differences in airways between males and females also affect lung function. Higher lung function was noted for males due to larger airways [62]. Respiratory muscle strength is another marker of anatomic sex differences and can be assessed by measuring hand grip strength [63]. Hand grip strength has been positively correlated with FEV1 only in males, while bone mineral content had the strongest association with FEV1 in females [63]. These findings underscore the role of anatomic sex differences between males and females and their contribution to differences in lung function. 

### 4.2. Airway Metrics and Pulmonary Functions in COPD

The Multi-Ethnic Study of Atherosclerosis (MESA) is a population-based dataset initially designed for non-pulmonary diseases, but it has been probed to examine COPD-related questions [64]. Since MESA did not aim to recruit COPD-affected individuals, it provides unique opportunities to investigate the natural history of COPD. In a study by Oelsner et al., the square root of a 10 mm lumen perimeter (Pi10) and airway wall thickness (AWT) were shown to predict COPD risk in the MESA cohort [64]. Moreover, males were found to be in higher Pi10 quartiles than women, suggesting sex differences in pulmonary function decline related to the natural history of airway pathology [64]. Another MESA study by Armstrong et al. [65] addressed QTc-related arrhythmia in the context of pulmonary function with percentage of emphysema in the lungs. Even though sex was not found to be associated with emphysema, it was related to %FEV1 [65]. A study by Hoffman et al. [66] found that women from the MESA cohort had a lower percentage of emphysema. This study also addressed emphysema in different ethnic groups [66], another strength of the MESA study.

## 5. COPD Characteristics

### 5.1. COPD Prevalence

In the United States between 2011 and 2021, females were noted to have a higher incidence of COPD compared to males [67]. This trend varied internationally [68]. A recent Swedish study identified more females with COPD than males [31]. Other studies report a dramatic increase in the prevalence of COPD in female smokers compared to male smokers, suggesting increased susceptibility to cigarette smoke exposure in females [32,68]. Supporting this line of thinking, the cumulative smoke exposure for females diagnosed with COPD tends to be lower than for males [32,68]. Other studies emphasize a lower age of COPD incidence in females compared to males and a higher burden of symptoms and disease severity (Table 1) [31,32,33,34].

### 5.2. COPD Symptoms

Multiple studies report a higher dyspnea symptom burden and severity in females than males (Table 1) [33,34,35,36,37]. However, Guenette et al. [35] found that after accounting for either body mass or ventilatory capacity, the sex differences in breathlessness disappeared (Table 1) [35]. A higher frequency of severe dyspnea was reported in females when assessed by the modified Medical Research Council (mMRC) dyspnea questionnaire (Table 1) [33,35,37], and a higher severity of dyspnea during strenuous activity was observed in females even within the group of patients with mild COPD [35]. The burden of breathlessness assessed by the University of California, San Diego Shortness of Breath Questionnaire was higher in females than males at given ages, percent predicted values of FEV1%, emphysema proportions of the lungs quantified by chest CT, and cigarette exposure pack-years [38] and contributed to an overall worse mental and physical well-being [37]. Sputum production tends to be either lower or less frequently reported by females [39]. No sex differences were reported in the severity of coughing [33]. 

### 5.3. COPD Exacerbations and Comorbidities

Studies examining sex differences in AE-COPD consistently report more frequent AE-COPD in females than in males, even accounting for GOLD stage groups (Figure 1) [31,34,37,40,41,42]. Females with a diagnosis of COPD are reported to have a shorter median time to first AE-COPD than males within the frequent AE-COPD phenotype [41]. Despite the higher risk of exacerbations for females than males, no sex differences have been reported regarding the severity of AE-COPD [31,41]. AE-COPD is linked to comorbidities and mortality in COPD [69]. More prevalent comorbid conditions in males include hypertension, congestive heart failure, anemia, chronic ischemic heart disease, chronic heart failure, and gout [43,44]. In females with COPD, more prevalent comorbid conditions include stroke, asthma, osteoporosis, and psychological conditions such as anxiety and depression (Figure 1) [36,40,43]. The prevalence of these comorbidities differs based on self-reported ethnicity as well [70]. As an example, African American females with COPD were found to be at higher risk of cardiovascular disease-related mortality than non-Hispanic white females [70]. 

### 5.4. COPD Mortality

The most severe GOLD stages are characterized by a mortality rate 3.5 times higher than the healthy population [71]. There is, however, no consensus in the literature regarding sex differences in COPD mortality, with many studies reporting contradicting results [33,41,72]. One study reported no difference in mortality rate due to COPD but a higher all-cause mortality in males [31]. Notably, many studies report lower survival rates for males than females considering all-cause mortality, although these differences may arise due to differences in populations. Speculations related to higher mortality among males were in the context of higher historical disease prevalence in males [43]. Different prevalences of comorbid conditions between males and females also likely caused the difference in mortality unrelated to COPD, complicating the interpretation of the available and reported outcome data [31]. A recent study by Li et al. [44] using NHANES data revealed that higher mortality in males persisted even after accounting for certain comorbid conditions. Generally, reporting all-cause mortality rates for patients with COPD is simpler because it is challenging to ascertain respiratory vs. non-respiratory causes of mortality. Most studies to date also do not differentiate COPD subphenotypes while evaluating mortality differences between males and females. Regional medical resources and cultural perception between males and females vary significantly. Future studies could benefit from considering these variables to provide more generalizable, relevant findings. 

## 6. Sex Differences in Disease Management

### 6.1. Sex Differences in Non-Pharmacological Treatment

Regardless of sex, the most effective non-pharmacological treatment for COPD patients is smoking cessation. Interestingly, former female smokers were found to be more prone to start smoking again than males [73]. However, studies focused on sex differences in smoking cessation are conflicting because of confounding factors and variations in samples across location and time, impacting quit rates in females vs. males [73]. Regardless, reports of females having more difficulty with smoking cessation play an important role in devising treatment plans, as female smokers may experience more significant benefits with cessation [32,73]. A Swedish study by Åberg et al. [74] reported that females are more likely to receive smoking cessation support than males but take sick leave due to AE-COPD more frequently. Higher utilization of non-pharmacological management options in females was noted by Lutter et al. in a German cohort study [75]. Smoking cessation interventions were offered comparably between males and females, but females had a significantly higher chance of accepting the intervention [75]. A Swedish cohort study also reported similar findings [36]. Non-pharmacological treatment is an integral part of the COPD management plan. A study by Isgrò et al. noted that educational programs targeting the physicians impacted proposed therapies for males and females with COPD [76]. They concluded that COPD-related educational interventions in primary care settings reduced the sex disparities in the overall quality of care. In contrast, at baseline, there was a bias towards males receiving more therapeutic options [76]. Most of the studies highlight bias towards females with COPD receiving more non-pharmacological management options, but these studies revealed a sex bias favoring males at baseline. This observation raises a possibility that regional and cultural variations may impact clinical care for smoking cessation. The topic of non-pharmacological treatment of COPD still lacks insight from broad populations and is a potential research question for future studies.

### 6.2. Sex Differences in Pharmacological Treatment

Global trends in sex differences regarding pharmacological COPD treatment have not been well reported in COPD literature. Existing reports are limited to a few countries and tend to report contradictory results. In a Swedish study by Lisspers et al. [31], females with COPD were prescribed more inhaled corticosteroids (ICS) or a combination of ICS with long-acting beta-agonists (LABA) [31]. The authors of this study highlight co-existing asthma as a likely factor explaining this trend [31]. A study by Sundh et al. highlighted that the triple inhaled therapy (a combination of long-acting beta-agonist, long-acting anticholinergic, and inhaled steroid) was offered to females more frequently than males [77]. A different Swedish study by Åberg et al. reported that females were prescribed a triple inhaled therapy and more robust pharmacological treatment than males [74]. Alternatively, a recent study by Milne et al. found no association between sex and pharmacologic therapy for individuals with COPD [15]. A study of COPD management in Sicily identified males as more likely to receive a higher quality of care than females [76]. Some have speculated that the observed sex differences in COPD management may be caused by sex bias. A cross-sectional study conducted in the USA by Rinne et al. reported a large proportion of female veterans with COPD being administered inappropriate and less inhaler therapies before hospitalization than males [78]. The limitations of this study, however, include no smoking status and missing clinical respiratory symptoms, which may have influenced clinical decision-making [78]. This study also did not detect sex differences in hospital readmission rates [78]. A study by Isgrò et al. concluded that divergent healthcare settings and socioeconomic factors between countries heavily impact the sex differences in COPD management [76]. A study by Akbarshahi et al. disputed such bias in diagnostic approaches and therapies [79]. Still, a study by Mamary et al. of the US population showed that males are more frequently underdiagnosed with COPD [80]. The differences among these studies are hypothesized to be related to regional variations in clinical practice. The Swedish studies by Lisspers et al. [31], Sundh et al. [77], Aberg et al. [74] all highlight females receiving more pharmacological treatment for COPD, while the studies by Milne et al. [15], Isgrò et al. [76], and Rinne et al. [78] either report no sex differences or contradict. This emphasizes that study locations and sociodemographic factors can confound reported sex disparities and are heavily influenced by regional clinical practice patterns and local healthcare finance. All of these observations highlight the need for a more comprehensive study to address sex differences in pharmacological and non-pharmacological COPD management accurately. This will help to identify the factors and therapeutic approaches that can reduce gaps in COPD management caused by sex differences and improve management strategies to address these discrepancies.

## 7. Conclusions

This review explored sex differences in COPD related to disease characteristics, phenotypes (subphenotypes), risk factors impacting disease development, and non-pharmacologic and pharmacologic disease management. In general, many studies suffer from confounding variables such as geography, regional culture, and financial differences, in addition to study designs that fail to consider sex differences adequately. Emerging reports highlight that autosomal, epigenetic, and sex chromosomal genetics interact with sex hormones and environmental factors during the natural history of COPD. Based on the results of this review, we recommend the following to be considered for future investigations. First, careful consideration for geographic, cultural, and financial variabilities will help increase scientific rigor and generalizability. Second, some COPD risk factors show a strong tendency to stratify by sex differences. These observations suggest focusing on higher-risk groups, such as developing better indoor ventilation as an intervention arm of a clinical trial for non-smoking female heavily exposed to indoor biomass smoke. Third, our review demonstrates a strong possibility that sex differences drive unique COPD pathology. Therefore, all clinical and preclinical studies must have a sex-balanced design while considering autosomal and sex chromosomes, epigenetics, and sex hormones. These requirements would likely increase the burden on the study design and resources but are necessary to maintain scientific rigor. Fourth, more clinical trials are required to determine how clinical management can be tailored based on sex differences. Additional research on the interactions of sex and treatment could give a unique opportunity to personalize the individual treatment plan and improve the morbidity and mortality of COPD patients.

## Figures and Tables

**Figure 1 ijms-26-02747-f001:**
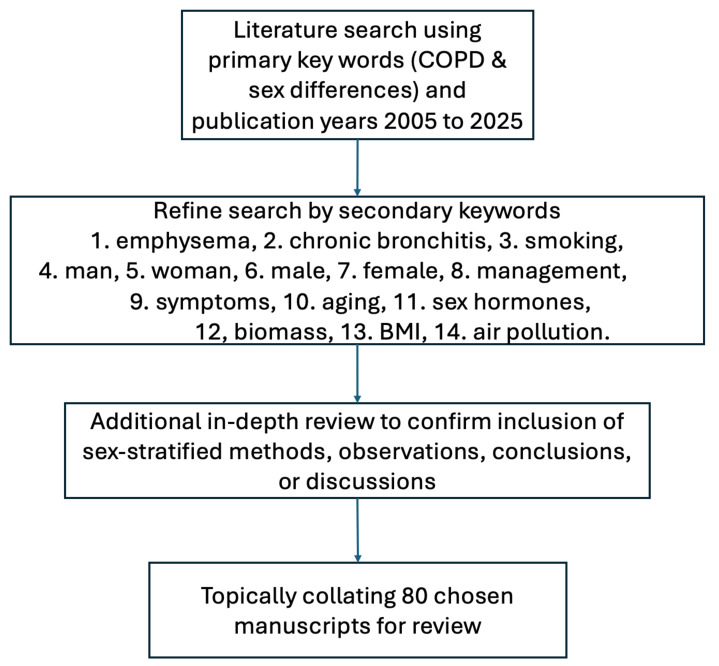
Overview of manuscript selection.

**Table 1 ijms-26-02747-t001:** Summary of reported sex differences in disease characteristics and risk factors.

Risk Factors	
Cigarette Smoking	Females are more susceptible to harmful effects of **cigarette smoke** than males due to anatomic vulnerabilities [21,22].
Biomass Fuel Exposure (Domestic)	Females face higher exposure to **biomass smoke** during household cooking and chores with a lack of proper ventilation [23,24]. **Biomass fuel exposure** in India [22] and China [24] is strongly associated with COPD incidence among females, especially in rural and developing regions. Females exposed to **biomass smoke** were also found to be at much higher risk of developing COPD than their male counterparts [24].
Industrial Exposure (Occupational)	Certain branches of **industry** and **occupations** have higher COPD incidence in one sex than the other [25].
Air Pollution (Ambient)	No consensus in terms of which sex is more susceptible to **air-pollution**-related AE-COPD [26,27].
BMI	Among **underweight** individuals, there was a higher prevalence of COPD in males than in (underweight) females [28].
Aging	The absolute decline in FEV1 per year is more pronounced for males than females [29] and the COPD prevalence is higher with more **advanced ages** and more frequent in males [30].
**Disease characteristics**	
Phenotypic difference	Most studies report **emphysema** as more prevalent and more severe in men [12,13,14,15]; however, there is a lack of consensus on the prevalence of **chronic bronchitis** in either sex [17,18,19,20].
Prevalence	Women were reported as being more likely to develop COPD at a **younger age** than men [31,32,33].
Symptoms	More frequent and severe **dyspnea** with lower **sputum production** for women [33,34,35,36,37,38,39], who also reported a lower **quality of life** [37].
Exacerbations	Women diagnosed with COPD are at higher risk of **exacerbations** than men from the same age group [31,34,37,40,41,42].
Comorbidities	**Comorbidities** differ by sex, and some are more frequent in one of them; however, studies consistently report more frequent psychological **comorbidities** in women [36,40,41,43,44].

## Data Availability

No new data were produced in this manuscript.

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
