# Peer review of "Sex Differences in Chronic Obstructive Pulmonary Disease: Implications for Pathogenesis, Diagnosis, and Treatment"

_ijms, 2025, doi:10.3390/ijms26062747_

Round 1
Reviewer 1 Report
Comments and Suggestions for Authors
This is an interesting article that reviews biological sex in COPD development and other implications. However, in my opinion, the article could be improved by clarifying some aspects of the impact of sex differences on this pathology.
1. COPD phenotypes. The authors identify main phenotypes inn COPD: emphysema and chronic bronchitis, the latter being identified by a clinical definition of cough and sputum production. However, from an anatomical and histological point of view the prevalence of a centrolobular and panlobular emphysema is quite clear; the centrilobular being characterized, mainly by the involvement of bronchial alterations and thickening of bronchial walls, and the panlobular by alveolar damage (see Hogg JC). I think that this should be mentioned in the presentation of the principal COPD phenotypes and, consequently, it should be indicated if there are sex differences between centrolobular and panlobular emphysema prevalence.
2. COPD risk factors: cigarette smoking. Do females and males of similar body deimension (weight, etc. etc.) and similar smoking exposure (pack/year) have a similar degree of COPD disease or emphysema? Please, clarify this point.
3. COPD risk: ageing and associated physiological changes. In this paragraph you report that “Females are reported to experience a more rapid lung function decline than males (37)”. However, it is not clear if females and males are compared normalizing lung functional measurements and pack/year and patient size. A critical comment of the cited articles (references 37, 38, 39) should be reported in the text.
Author Response
Reviewer 1: Comments and Suggestions for Authors
This is an interesting article that reviews biological sex in COPD development and other implications. However, in my opinion, the article could be improved by clarifying some aspects of the impact of sex differences on this pathology.
- COPD phenotypes. The authors identify the main phenotypes in COPD: emphysema and chronic bronchitis, the latter being identified by a clinical definition of cough and sputum production. However, from an anatomical and histological point of view the prevalence of a centrolobular and panlobular emphysema is quite clear; the centrilobular being characterized, mainly by the involvement of bronchial alterations and thickening of bronchial walls, and the panlobular by alveolar damage (see Hogg JC). I think that this should be mentioned in the presentation of the principal COPD phenotypes and, consequently, it should be indicated if there are sex differences between centrolobular and panlobular emphysema prevalence.
Response: We thank the reviewer for this insightful comment about the sub-phenotypes of the emphysematous. We agree that essential differences in centrilobular vs. panlobular emphysema are worth describing in our review. We have added this information in Page 3.
- COPD risk factors: cigarette smoking. Do females and males of similar body dimension (weight, etc. etc.) and similar smoking exposure (pack/year) have a similar degree of COPD disease or emphysema? Please, clarify this point.
Response: We thank the reviewer for raising this relevant point regarding our review of COPD risk factors. Please see clarifying statements in Page 4.
- COPD risk: ageing and associated physiological changes. In this paragraph you report that “Females are reported to experience a more rapid lung function decline than males (37)”. However, it is not clear if females and males are compared normalizing lung functional measurements and pack/year and patient size. A critical comment of the cited articles (references 37, 38, 39) should be reported in the text.
Response: We agree with the reviewer that this portion of the manuscript required clarification. We have now expanded upon these cited references to provide additional details in Page 6.
Reviewer 2 Report
Comments and Suggestions for Authors
The article “Sex Differences in Chronic Obstructive Pulmonary Disease: Implications for Pathogenesis, Diagnosis, and Treatment” provides an insightful overview of the existing literature on sex-based differences in COPD. However, there is a lack of a clear and structured methodology for selecting and analysing the data included in the review, which raises some concerns about its transparency and reliability.
The article is presented as a review, but it is not stated whether it is a narrative review, an integrative review or a systematic review. This omission makes it difficult to determine the rigour of the data selection process. A robust review will usually include a methodology section describing:
· The databases searched (e.g. PubMed, Scopus or Web of Science);
· The search terms or keywords used;
· The criteria for inclusion and exclusion of studies;
· The time frame of the studies considered.
Without this information, the reader cannot judge whether the review covers the topic comprehensively or is focussed on specific sources or regions.
While the article refers to numerous studies, it is not clear how these studies were identified or prioritised.
The article cites studies from various regions, including the United States, Sweden and China, but does not address how regional differences in health care systems, environmental exposures or sociocultural factors might influence the results.
It also emphasises certain risk factors, such as exposure to biomass fuels, in regions such as India and China, while they may not be as relevant in industrialised countries. Without a discussion of these regional correlations, the generalisability of the results remains unclear. The table is an important summary, which must always be included in the text. Furthermore, the authors have limited themselves to a single bibliographical reference in point 1 of the text and have thus failed to provide a comprehensive analysis. A review article should summarize the results of multiple studies to ensure that the conclusions are representative of the broader scientific consensus. Relying on a single source undermines the reliability of the discussion and raises concerns about possible bias or oversimplification.
In summary, while the article provides valuable insights into sex-based differences in COPD, its reliability and reproducibility is limited by the lack of methodological detail. The inclusion of a more structured and transparent methodology would greatly enhance the credibility and impact of this study. By addressing these issues, future work can provide a more solid foundation for understanding and addressing gender differences in the pathogenesis, diagnosis and treatment of COPD.
addressing gender differences in the pathogenesis, diagnosis and treatment of COPD.
Comments on the Quality of English Language
The English could be improved to more clearly express the research.
Author Response
Reviewer 2: Comments and Suggestions for Authors
The article “Sex Differences in Chronic Obstructive Pulmonary Disease: Implications for Pathogenesis, Diagnosis, and Treatment” provides an insightful overview of the existing literature on sex-based differences in COPD. However, there is a lack of a clear and structured methodology for selecting and analysing the data included in the review, which raises some concerns about its transparency and reliability.
- The article is presented as a review, but it is not stated whether it is a narrative review, an integrative review or a systematic review. This omission makes it difficult to determine the rigour of the data selection process. A robust review will usually include a methodology section. Without this information, the reader cannot judge whether the review covers the topic comprehensively or is focused on specific sources or regions.
While the article refers to numerous studies, it is not clear how these studies were identified or prioritised.
Response: We thank the reviewer for these comments regarding our review methodology. To address this comment, we revised and added the Methods section, which details our approach (a new Figure 1) to the literature search and the inclusion of relevant references in our review.
- The article cites studies from various regions, including the United States, Sweden and China, but does not address how regional differences in health care systems, environmental exposures or sociocultural factors might influence the results. It also emphasises certain risk factors, such as exposure to biomass fuels, in regions such as India and China, while they may not be as relevant in industrialised countries. Without a discussion of these regional correlations, the generalisability of the results remains unclear.
Response: While the reviewer’s point is salient, the requested generalizability of some of these risk factors is neither achievable nor realistic based on the results of our review. Our purpose in presenting regional risks such as biomass is to alert the readers from these regions with low prevalence. We believe that heightened awareness is critical, especially in the current international migrant crisis, which brings populations from these at-risk regions to developed countries. We edited our manuscript to highlight the difficulty of generalizing this topic and the critical nature of these observations.
- The table is an important summary, which must always be included in the text. Furthermore, the authors have limited themselves to a single bibliographical reference in point 1 of the text and have thus failed to provide a comprehensive analysis. A review article should summarize the results of multiple studies to ensure that the conclusions are representative of the broader scientific consensus. Relying on a single source undermines the reliability of the discussion and raises concerns about possible bias or oversimplification.
Response: We thank the reviewer for recognizing the value of the table we presented in our manuscript. Regarding the single bibliographic reference for the first point in our manuscript, we would like to clarify our rationale for the single reference in the Background section. This portion of the text is intended as background, and the reference cited (“2024 GOLD Report”) provides the gold standard clinical definition of COPD in the field. This one reference includes more than 100 references of its own to define COPD. Therefore, we believe this one reference is sufficient to support the purpose of that section. To clarify the purpose of this section in the manuscript, we have renamed it to “Background: COPD – Definition, Symptoms, Diagnosis.”
Round 2
Reviewer 2 Report
Comments and Suggestions for Authors The authors have gone to great effort to improve the manuscript, and I appreciate the revisions. However, I still believe that it would be important to increase the number of references in section 1. I recognise the authors' rationale that this part of the text serves to provide a historical overview and that the cited reference (“2024 GOLD Report”) represents the gold standard of clinical definition of COPD in the field. Furthermore, this reference contains over 100 citations of its own that support the definition of COPD. While I can understand this reasoning, I still believe that the inclusion of additional references would strengthen this section. Comments on the Quality of English Languagemust to be improve
Author Response
- I recognise the authors' rationale that this part of the text serves to provide a historicaloverview and that the cited reference (“2024 GOLD Report”) represents the gold standard ofclinical definition of COPD in the field. Furthermore, this reference contains over 100 citations ofits own that support the definition of COPD. While I can understand this reasoning, I still believethat the inclusion of additional references would strengthen this section.
Response: We'd like to thank the reviewer's feedback to ensure appropriate references are cited. We have now edited this introduction section and added seven more references. Comment highlights the section in which additional references can be found.